# Combined Effect of Cold Atmospheric Plasma and Chitooligosaccharide–EGCG Conjugate on Quality and Shelf-Life of Depurated Asian Green Mussel

**DOI:** 10.3390/foods14081399

**Published:** 2025-04-17

**Authors:** Ajay Mittal, Soottawat Benjakul, Nigel Brunton, Deepak Kadam, Avtar Singh

**Affiliations:** 1UCD Institute of Food and Health, University College Dublin, Belfield Campus, D04 V1W8 Dublin, Ireland; ajay.mittal@ucd.ie (A.M.); nigel.brunton@ucd.ie (N.B.); 2International Center of Excellence in Seafood Science and Innovation (ICE-SSI), Faculty of Agro-Industry, Prince of Songkla University, Hat Yai 90110, Songkhla, Thailand; soottawat.b@psu.ac.th; 3Department of Food and Nutrition, Kyung Hee University, Seoul 02447, Republic of Korea; 4Department of Food and Human Nutritional Sciences, University of Manitoba, Winnipeg, MB R3T 2N2, Canada; deepak.kadam@umanitoba.ca

**Keywords:** chitooligosaccharide-EGCG conjugate, Asian green mussel, cold plasma, shelf-life extension, food safety

## Abstract

The combined effects of chitooligosaccharide–epigallocatechin gallate conjugate (CEC) at different concentrations (1, 2, and 3%, *w*/*w*) and cold atmospheric plasma (CAP) on the depurated Asian green mussel edible portion (AGM-EP) were investigated during refrigerated storage for 15 days. Among all the treatments, the microbial counts, total volatile bases (TMA-N and TVB-N), and lipid oxidation of AGM-EP-treated 3% CEC in conjunction with CAP (CEC-3-CAP) were lower than the other samples during 15-day storage (*p* < 0.05). Total viable bacteria (6.16 log CFU/g sample), psychrotrophic bacteria (3.24 log CFU/g sample), *Vibrio* spp. (2.47 log CFU/g sample), presumptive *Pseudomonas* (5.93 log CFU/g sample), and H_2_S-producing bacteria (5.05 log CFU/g sample) counts of the CEC-3-CAP were lower than samples treated with 1 and 2% (*w*/*w*) CEC on day 15, as well as samples solely treated using CAP during refrigerated storage, irrespective of storage time. Additionally, CEC-3-CAP had significantly lower lipid oxidation (PV: 8.36 mg cumene hydroperoxide/kg sample and TBARS: 2.65 mg MDA/kg sample) as compared to those without CEC added and other samples (*p* < 0.05). The incorporation of CEC effectively mitigated lipid oxidation as supported by lower reduction of PUFAs in AGM-EP. Moreover, on day 0, no significant differences were observed in cooking loss or textural parameters (firmness and toughness) among the treatments (*p* > 0.05). However, as storage progressed, cooking loss increased in the CEC-3-CAP sample, while a noticeable decline in firmness and toughness was recorded (*p* < 0.05). This further attributed to the lower likeness attained for CAP-3-CAP on day 12, but the score was higher than the acceptable limit (5.0). Therefore, CAP together with CEC is a promising technology to prolong the shelf-life of depurated AGM-EP by at least 9 days as compared to the control (3 days), but it certainly needs further studies for the retention of textural properties and sensorial attributes.

## 1. Introduction

Asian green mussel (AGM; *Perna viridis*) is a species of the family Mytilidae of commercial importance in China, Taiwan, and Southeast Asian countries such as Myanmar, Thailand, Singapore, Indonesia, and the Philippines to fisheries and aquaculture because of its rapid growth rate and continuous breeding. In 2021, Southeast Asian countries reported a total AGM production of 106,201 MT [1]. Thailand accounted for approximately 50% of this output, producing 52,247 MT, valued at USD 18.15 million [1]. With a protein content of 36% [2], mussel is a cheap source of protein for human consumption. Apart from protein, mussel meat contains carbohydrate, lipids, and minerals and vitamins [3]. In general, AGM had 79–80% moisture, 13–14% protein, 1–2% fat, and 1–2.5% ash on a wet basis as reported by Sorio and Inolino [4]. Despite all the benefits associated with the intake of seafood, there are challenges facing the industry. Seafood quality starts to deteriorate post catch and during storage due to a high moisture content, which allows the proliferation of spoilage bacteria and other factors such as endogenous proteases [5,6]. The microorganisms produce various enzymes, such as decarboxylases, which degrade amino acids and proteins and can alter the flavors of the seafood. Moreover, microbial metabolism releases various volatile compounds, which negatively affect the odor of seafood products. Thailand is a tropical country with a warm climate, in which AGM loses its quality and texture within 1–2 days, thus lowering its market value. Moreover, AGM has its habitat in marine waters and concentrates microorganisms from surrounding waters during the filter-feeding process, therefore pathogenic bacteria such as *Vibrio* spp. are easily accumulated. For instance, Palamae et al. [7] and Sharma et al. [6] identified *V. parahaemolyticus* that possessed *tdh* and *trh* genes from AGM collected from different locations in Thailand. Seafood contaminated with *V. parahaemolyticus* can have serious implications in terms of gastroenteritis, cholera, wound infection, and primary septicemia when consumed [8]. Therefore, to ensure their microbial safety and overall quality, bivalves are commonly subjected to “depuration”, wherein they are held in clean freshwater to eliminate accumulated contaminants and endogenous microbiota prior to consumption [9,10]. Chinnadurai et al. [11] found that the complete elimination of *Vibrio* spp. from AGM was achieved after depuration using seawater for 24 h. Furthermore, instead of water, depuration using chitooligosaccharide–epigallocatechin gallate conjugate (CEC) solution reported to enhance the shelf-life of AGMs by 4 days [5]. Moreover, further appropriate treatment of AGMs using natural additive and non-thermal processing (NTP) technology would extend their shelf-life while preserving their texture.

In recent years, the food industry has shown growing interest in innovative processing techniques that target microbial inactivation without compromising product quality. NTP technologies, such as high-pressure processing, pulsed electric fields, ultrasound, and, more recently, cold atmospheric plasma (CAP), have gained considerable attention. These technologies function at ambient or moderately elevated temperatures (typically below 50 °C), thereby reducing the detrimental effects associated with conventional thermal processing, while effectively inactivating a broad spectrum of microorganisms, including bacteria, fungi, spores, viruses, and biofilms [12,13]. Among these, CAP has gained attention due to its ability to generate a broad range of reactive species, such as ozone, nitrogen and carbon oxides, free radicals, and charged ions, depending on the gases used in the process [14]. CAP has also demonstrated effectiveness in prolonging the shelf-life of meat products [15], including seafood like dried fish and fish fillets [16], mackerel [17], and Asian sea bass [18,19]. The reactive species produced during plasma treatment are responsible for microbial inactivation. However, they may also trigger chemical and biochemical alterations in food components, potentially compromising product quality [20]. Several studies have indicated that lipid oxidation rates in CAP-treated seafood remain significantly higher than in untreated samples even after several days of storage [21]. For instance, Shen et al. [22] reported an increase in thiobarbituric acid reactive substance (TBARS) level in dried fish samples, rising from 1.12 to 1.41 and 1.96 mg/kg following CAP treatment for 3 and 5 min, respectively. Similarly, Wang et al. [23] observed higher TBARS levels in CAP-treated tilapia. Given these concerns, researchers have speculated that incorporating antioxidants may help mitigate CAP’s adverse effects.

Shrimp shell chitooligosaccharide (CHOS) has been widely studied for its strong antioxidant and antimicrobial properties [24], along with various health-related benefits. CHOS has proven effective in preserving seafood products such as fish fillets and shrimp, particularly when combined with polyphenols [25]. Singh and Benjakul [26] found that Asian sea bass slices treated with both CAP and CHOS (at concentrations of 0.1% or 0.2%, *w*/*w*) exhibited reductions in TBARS level and peroxide value (PV) by 28–64% and 40–46%, respectively, compared to samples treated with CAP alone over 18 days of refrigerated storage at 4 °C. Furthermore, modifying CHOS into phenolic conjugates, such as CEC, has been shown to enhance its antioxidant and antimicrobial potential [27]. CEC has exhibited potent antibacterial activity, likely due to its ability to disrupt biofilm formation, impair bacterial motility, and induce severe cell wall damage, ultimately leading to protein leakage and DNA binding [6]. The application of CEC for extending the shelf-life of the depurated AGM edible portion (AGM-EP), in combination with CAP, presents a promising natural alternative to address concerns over antimicrobial resistance. In addition, CEC has exhibited digestive enzymes (amylase and glucosidase) inhibition and anti-obesity and anti-hypertensive (ACE and renin inhibition) properties [28,29]. Thus, it could provide health benefits to the consumers.

This study aimed to evaluate the combined effect of CEC and CAP on the physicochemical and preliminary sensory properties of the AGM-EP during 15 days of storage at 4 °C. The rationale for exploring this combination stems from the need to mitigate the adverse effects such as lipid oxidation that can be induced by CAP. By incorporating a natural preservative, CEC, this study aims to counteract these negative impacts while also addressing concerns related to antimicrobial resistance. This approach seeks to enhance the overall stability and quality of the AGM-EP, offering a more sustainable solution for food preservation.

## 2. Materials and Methods

### 2.1. Chemicals, Microbiological Media, and Preparation of Chitooligosaccharide–EGCG Conjugate (CEC)

All chemicals were purchased from Sigma-Aldrich (St. Louis, MO, USA). Microbial media were procured from Oxoid (Thermo Fisher Scientific, Waltham, MA, USA). CHOS and CEC was prepared as per our previous methods [27].

### 2.2. Collection and Extraction of Edible Portion of Asian Green Mussel

Live AGMs (46.3 ± 2.9 g per piece; *n* = 100) in the adult stage were sourced from a seafood market in Songkhla province on the same day, Thailand, and transported to the laboratory within 1 h in bags packed with crushed ice. AGMs were morphologically characterized as described by Carpenter and Niem [30]. AGM has a smooth and elongate shell, with a distinctive downward-pointing beak and pearly white to iridescent blue internal shell valves. The anterior adductor scar is absent in adults. Upon arrival, AGM shells were thoroughly cleaned using a brush, rinsed with tap water, and drained. Thereafter, the AGMs underwent depuration in CEC solution (2%, *w*/*v*) for 3 h at room temperature as following previous study [5]. After depuration, AGMs were placed in polyethylene/polyamide bags, vacuum sealed, and cooked by immersion in boiling water for 1 min. AGMs were immediately transferred to iced water for rapid cooling. Finally, the edible portion from AGMs was hand-shucked using a sterilized knife for further treatment.

### 2.3. CEC and CAP Treatment of Asian Green Mussel 

CEC was first dissolved in a minimal amount of water (2.5 mL per 100 g of AGM-EP) at concentrations of 1, 2, and 3% (*w*/*w*). The solution was then manually mixed with AGM-EP. Thereafter, the prepared samples (25 g) were packed in a multilayer low-density polyethylene bag (18 × 28 cm^2^; thickness: 80 µm; oxygen permeability: 47.62 cm^3^/m^2^/day at 38 °C) along with a working gas mixture of 90%Ar and 10%O_2_ at a 3:1 (*v*/*w*) ratio using a Henkovac type 1000 (Tecnovac, Grassobbio BG, Italy). This working gas ratio yielded a higher antimicrobial impact and lower lipid/protein oxidation, as optimized in our previous work [31]. Each sealed bag was placed between the two electrodes and subjected to CAP treatment at 16 kV RMS for 5 min at room temperature. A control sample without CEC and CAP treatment was labeled as CON-DP, while a second control sample, treated with CAP without CEC, was designated as CON-DP-CAP. The samples treated with CEC at 1, 2, and 3% in combination with CAP was named CEC-1-CAP, CEC-2-CAP, and CEC-3-CAP, respectively. All the samples were stored at 4 °C for 15 days and analyzed on every 3rd of the storage under refrigeration having humidity of 38 ± 0.02%.

### 2.4. Microbiological and Chemical Analyses

#### 2.4.1. Microbial Counts

Samples of AGM-EP (25 g each) were homogenized and analyzed for microbial content. Total viable bacterial count (TVBC) was determined using the spread plate method on plate count agar [32]. Psychrophilic bacteria count (PBC) and presumptive *Pseudomonas* spp. count (PPC) was enumerated as per the methods of ISO 17410 [33] and ISO 13720 [34], respectively. An enumeration of H₂S-producing bacteria (HSPB) and *Vibrio* spp. was conducted using triple sugar iron agar and thiosulfate citrate bile salts sucrose agar, respectively. The plates for psychrophilic bacteria were incubated at 4 °C for 7 to 10 days, whereas all the other samples were incubated at 37 °C for 18 to 24 h. Results were reported as log CFU/g sample.

#### 2.4.2. Total Volatile Base Nitrogen (TVB-N) and Trimethylamine Nitrogen (TMA-N), Pexoide Value (PV), and Thiobarbituric Acid Reactive Substance (TBARS) Content

TVB-N and TMA-N content were determined using the Conway disc method as described by Prabhakar et al. [35]. PV and TBARS were determined to determine the extent of lipid oxidation as per the methods of Richards and Hultin [36] and Olatunde et al. [31], respectively.

#### 2.4.3. Texture Analysis

The toughness and firmness of the middle portion of the AGM-EP (*n* = 10) were analyzed using a TA-XT2 texture analyzer (Stable Micro Systems, Surrey, UK) using a Warner-Bratzler blade (speed rate of 2 mm/s) [37].

#### 2.4.4. Cooking Loss

The cooking loss of the sample was measured from the difference of weight before and after steaming for 5 min when a core temperature of 85 °C was obtained [37]. Cooking loss was calculated as follows:(1)$$Cooking\;loss\;\left(\%\right)=\left[\frac{A-B}{A}\right]\times 100$$
where A is the initial weight before steaming, and B is the weight after cooking, cooling, and draining.

#### 2.4.5. Fatty Acids Profile

The fatty acids composition was analyzed from fatty acid methyl esters (FAMEs), which were prepared from lipid extracts obtained from AGM meat using the Bligh and Dyer extraction method [38]. FAMEs dissolved in hexane were injected into a CPSil88 column (100 m × 0.25 mm, df = 0.2 μm) with a split ratio of 1/20. Gass chromatography (GC) analysis was carried out using the following conditions: injection temperature of 250 °C, detector (FID) temperature of 270 °C, column temperature of 200 °C, gas injection pressure of 31.62 psi, and helium as the carrier gas (constant flow 1.0 mL/min) [39].

#### 2.4.6. Sensory Evaluation

A total of 50 panelists (30 males and 20 females, aged between 25 and 40 years) participated in the study. They were instructed to rate the appearance, color, texture, odor, taste, flavor, and overall acceptability of the boiled (10 min) AGM-EP. A 9-point hedonic scale was used to provide the scores, as suggested by [40]. To ensure food safety, only samples with microbial counts below the permissible limit of 6 log CFU/g sample were included in the evaluation. A score below 5 on the scale was considered indicative of an unacceptable product [40].

### 2.5. Statistical Analyses

All experiments, except sensory evaluation, followed a completely randomized design. Triplicate measurements were used for all tests. One-way and two-way ANOVAs were performed, and Tukey’s test was applied for post hoc comparisons using IBM SPSS Statistics 29.0 authorized from University College Dublin, Dublin, Ireland. Sensory analysis data were analyzed using the Friedman test, and the Wilcoxon–Bonferroni test was used for pairwise comparisons following the Friedman test, using IBM SPSS Statistics 29.0 authorized from University College Dublin, Dublin, Ireland. Significance was set at *p* ≤ 0.05.

## 3. Results and Discussion

### 3.1. Changes in the Microbial Counts

TVBC, PBC, PPC, *Vibrio* spp. count (VC), and HSPB count (HSPBC) of all the samples are given in Figure 1. On day 0, TVBC for all the samples varied between 3.92 and 4.35 log CFU/g sample (Figure 1A). TVBC of all the samples increased as the storage time progressed. Among them, CON-DP exhibited the most significant rise, while CEC-3-CAP maintained the lowest count throughout the refrigerated storage period. TVBC is associated with the growth of mesophilic bacteria, which thrive within a moderate temperature range of 20–45 °C. These microorganisms are frequently present in seafood and can contribute to spoilage and potential foodborne illnesses. On day 3, the TVBC of CON-DP rapidly increased to 5.23 log CFU/g sample, which was higher than CON-DP-CAP and the other remaining samples (*p* < 0.05). The lower TVBC of CON-DP-CAP was due to the antimicrobial activity of ozone and other reactive species such as reactive oxygen species and reactive nitrogen species generated from CAP [41]. Among all the samples treated with CEC followed by CAP, CEC-3-CAP consistently demonstrated the lowest TVBC throughout the storage period (*p* < 0.05), highlighting the concentration-dependent antimicrobial potential of CEC. This effectiveness is likely due to the ability of CEC to disrupt microbial cell membranes and interfere with the structure or function of nucleic acids, such as DNA and RNA, leading to microbial cell damage or death [5,6]. In general, the hydroxyl and amino groups present in CEC are responsible for the antimicrobial activity. Similarly, Singh and Benjakul [26] reported a reduction in the TVBC and *Pseudomonas aeruginosa* of Asian sea bass slices when treated with CHOS or CAP. Buatong et al. [5] also noticed a reduction in TVBC in a dose-dependent manner when AGMs were depurated using CEC during refrigerated storage.

PBC of all the samples ranged between 3.42 and 4.21 log CFU/g sample on day 0 (Figure 1B). Similar to the TVBC, PBC in all the samples had steadily increased by day 3, and both control samples (CON-DP and CON-DP-CAP) demonstrated higher counts (5.17 and 4.40 log CFU/g sample, respectively) compared to the other samples (*p* < 0.05). CON-DP-CAP had a PBC lower than CON-DP (*p* < 0.05), irrespective of storage time, due to the antimicrobial action of CAP. Moreover, the synergistic effect of CEC treatment in combination with CAP in inhibiting psychrophile growth is also confirmed. AGM-EP treated with CEC in combination with CAP had a lower PBC than both control samples on day 3, irrespective of CEC concentration and storage time (*p* < 0.05). PBC was inversely related to CEC concentration, irrespective of storage time (*p* < 0.05). Psychrotrophic bacteria—notably *Pseudomonas, Shewanella,* and *Photobacterium*—are commonly recognized as the primary spoilage organisms in seafood stored under refrigerated conditions [42]. By day 6 of storage, the differences in PBC among treatments became more evident. Specifically, control samples (CON-DP and CON-DP-CAP) showed elevated PBCs of 5.43 and 5.02 log CFU/g sample, respectively. In contrast, the remaining treated samples exhibited lower bacterial loads, ranging from 3.95 to 4.41 log CFU/g sample, indicating improved microbial stability. Among CEC-treated samples, CEC-3-CAP had the lowest PBC, followed by CEC-2-CAP and CEC-1-CAP, respectively, on day 6 (*p* < 0.05). Similar trends were continued on days 9, 12, and 15.

VC, PPC, and HSPBC of the control samples (CON-DP and CON-DP-CAP) in comparison with samples treated with CEC in combination with CAP are illustrated in Figure 1C–E, respectively. VC of all the samples ranged between 4.83 and 5.31 log CFU/g sample on day 0, where VC of AGM-EP treated with CEC followed by CAP was not varied (*p* > 0.05) (Figure 1C). Vibrios such as *V. parahaemolyticus*, *V. campbellii*, *V. vulnificus*, *V. alginolyticus*, *V. harveyi*, *V. cholerae*, etc., are also the major causes of seafood-borne illnesses in terms of gastroenteritis, cholera, wound infection, and primary septicemia [43].

As storage progressed to day 3, unlike the TVBC and PBC, VC of all samples decreased and had a range between 3.36 and 4.10 log CFU/g sample, in which CON-DP had the highest VC, while CEC-3-CAP showed the lowest VC (*p* < 0.05). The decreasing trend was most likely linked to the synergetic antibacterial impact of CEC and CAP. The result was also in agreement with the TVBC on day 3. Similarly, Buatong et al. [5] also observed an reduction in VC when AGMs were depurated using CEC at 1 and 2% during storage for 6 days in refrigerated conditions. Palamae et al. [37] also reported a decrease in *Vibrio* spp. in AGM during refrigerated storage when treated with CHOS–catechin conjugate. CHOS–polyphenol conjugate inhibited *Vibrio* spp. growth through cell membrane damage, which leaked extracellular and intracellular content [36]. Additionally, it significantly suppressed biofilm formation and reduced bacterial motility. A decline in VC across all the samples was also observed on day 6 of storage. Nevertheless, irrespective of CEC concentration, the samples treated with CEC followed by CAP had a lower VC than other samples (*p* < 0.05), plausibly related to the effective synergistic effect of CEC and CAP. From day 9 to 15, VC of all samples steadily decreased (*p* < 0.05), and CEC-3-CAP possessed the lowest VC compared to the other samples (*p* < 0.05). The decrease in VC may be linked to nutrient depletion associated with the increased activity of spoilage microorganisms, as reflected in the rising TVC and PBC [4]. Moreover, *Vibrio* spp. typically flourishes in saline conditions, and the absence of such an environment might have further inhibited their proliferation [36].

On day 0, PPC levels in all the samples ranged from 3.82 to 4.37 log CFU/g sample (Figure 1D). *Pseudomonas* spp., psychrotrophic and Gram-negative bacteria commonly present in fresh meat, are a well-known contributor to seafood spoilage [44]. They produce proteolytic and other hydrolytic enzymes that break down fats, proteins, and carbohydrates in to aldehydes, ketones, hydrogen sulfide (H₂S), and esters, leading to the formation of slime and distinct sweet or fruity off-odors [45]. By day 3, CON-DP sample exhibited the highest PPC at 5.12 log CFU/g sample, significantly higher than other samples (*p* < 0.05), which had PPC values between 3.98 and 4.40 log CFU/g sample. Samples treated with a combination of CEC and CAP displayed lower PPC throughout storage, with CEC-3-CAP demonstrating the lowest PPC (*p* < 0.05). The presence of CAP appeared to enhance the antimicrobial effect of CEC, effectively suppressing the proliferation of *Pseudomonas* spp. Similar to TVC and PBC, PPC in all samples increased over time (*p* < 0.05), with CEC-3-CAP maintaining the lowest PPC and CON-DP the highest (*p* < 0.05). These findings suggest that the combined application of CEC and CAP contributed to a greater inactivation of *Pseudomonas* spp. during extended refrigerated storage. The same trend persisted throughout the storage period, with CEC-3-CAP consistently exhibiting the lowest PPC (*p* < 0.05).

On day 0, HSPBC ranged between 3.40 and 3.80 log CFU/g sample (Figure 1E), with no significant difference observed between CON-DP and CON-DP-CAP (*p* > 0.05). As storage progressed, HSPBC levels increased in all samples (*p* < 0.05), with the highest counts recorded in CON-DP, followed by CON-DP-CAP, CEC-1-CAP, CEC-2-CAP, and CEC-3-CAP, respectively. HSPB includes *Shewanella* spp., Gram-negative bacteria from the Shewanellaceae family, known for their ability to produce off-odors and hydrogen sulfide (H_2_S) through the breakdown of sulfhydryl compounds like cysteine and methionine [46]. By day 6, HSPBC had risen in all samples, with CON-DP and CON-DP-CAP exhibiting the highest HSPBC. However, the antimicrobial properties of CEC and CAP contributed to lower HSPBC in treated samples (*p* < 0.05). Notably, CON-DP-CAP had a reduced HSPBC compared to CON-DP due to the antimicrobial effect of CAP. Among all the samples, CEC-3-CAP consistently maintained the lowest HSPBC, followed by CEC-2-CAP, throughout the entire storage period (*p* < 0.05).

Overall, the bacterial inhibition was mainly associated with the antibacterial activity of reactive oxygen or nitrogen species and ozone produced by CAP and the synergetic effect of CEC, which not only reduced the bacterial count but also limited the negative impact of reactive species produced by CAP. For all the bacterial counts, similar trends were noticed in previous studies, where different technologies such as high pressure, acid electrolyzed treatment, sous vide, etc., were used to enhance the shelf-life of AGMs and hard clams [5,47,48].

### 3.2. Changes in TVB-N and TMA-N Contents

The levels of TVB-N and TMA-N serve as indicators of seafood quality and freshness [49]. Figure 2A,B illustrate the TVB-N and TMA-N content in AGM-EP samples treated with CEC followed by CAP, in comparison to control samples. On day 0, the TVB-N content across all the samples ranged from 6.68 to 9.25 mg N/100 g sample. Among them, CEC-3-CAP exhibited the lowest TVB-N content (*p* < 0.05). However, earlier studies by Palamae et al. [37] and Buatong et al. [5] reported lower TVB-N levels in the AGM-EP at the initial stage. Variations in TVB-N levels within the same species can be attributed to differences in origin, seasonality, and other intrinsic factors. As the storage time increased, TVB-N content of all the samples showed a significant rise (*p* < 0.05). By day 3, CON-DP and CON-DP-CAP samples had TVB-N values of 9.34 and 9.20 mg N/100 g sample, respectively (*p* > 0.05). Additionally, TVB-N content of CEC-treated samples combined with CAP remained statistically similar (*p* > 0.05) on day 3, except for CEC-3-CAP, which displayed the lowest value. This upward trend in TVB-N levels during refrigerated storage has been previously documented [5]. On day 6, no significant differences (*p* > 0.05) were observed between CEC-1-CAP and CEC-2-CAP or between CON-DP-CAP and CEC-3-CAP. However, by day 9, CEC-3-CAP recorded the lowest TVB-N content, followed by CON-DP-CAP, CEC-2-CAP, and CEC-1-CAP, respectively (*p* < 0.05). This pattern strongly correlated with the reduced total viable bacterial count (TVBC) and other microbial counts in CEC-3-CAP, suggesting a lower rate of deamination of non-protein nitrogenous compounds. Microorganisms responsible for spoilage contribute to the degradation of seafood products by generating volatile compounds such as ammonia, dimethylamine, and trimethylamine, collectively referred to as TVB [50]. Notably, CEC-3-CAP maintained the lowest TVB-N content on days 12 and 15, remaining below the permissible limit of 30 mg N/100 g sample. These findings suggest that the combination of CEC and CAP treatment effectively mitigated the increase in TVB-N levels in the AGM-EP throughout storage, likely due to its strong antimicrobial properties.

At the beginning of storage (days 0 and 3), TMA-N levels in all samples ranged from 0.93 to 0.97 mg N/100 g sample and increased to between 1.05 and 1.72 mg N/100 g sample (Figure 2B). The threshold for seafood freshness, based on TMA-N content, is 5 mg N/100 g sample [51]. These results confirm the freshness of the AGM used in this study. By day 6, the highest TMA-N concentrations (2.51 mg N/100 g sample) were observed in the CON-DP-CAP and CEC-1-CAP samples, showing a significant difference from the other groups (*p* < 0.05). The enzyme TMAO reductase, produced by spoilage bacteria, facilitates the conversion of TMAO to TMA, leading to undesirable odors and flavors [52,53]. By day 9 of extended storage, the CEC-3-CAP sample exhibited the lowest TMA-N level compared to the other samples (*p* < 0.05), a trend that persisted on days 12 and 15. This reduction was likely due to a lower microbial load, particularly HSPBC, in samples treated with a higher concentration of CEC in combination with CAP, effectively slowing down TMA-N formation in the refrigerated AGM-EP. Certain HSPB, including *Shewanella* spp., contribute to the fishy odor in stored seafood by converting TMAO to TMA through TMAO reductase activity [54,55]. Consequently, both TVB-N and TMA-N levels were directly linked to microbial load.

### 3.3. Changes in PV and TBARS Value

PV and TBARS level for the control samples (CON-DP and CON-DP-CAP) in comparison to the treated samples are illustrated in Figure 3A and Figure 3B, respectively. On day 0, PV across all the samples ranged from 5.75 to 6.09 mg cumene hydroperoxide equivalent/kg sample. Notably, CEC-3-CAP exhibited the lowest PV (*p* < 0.05; Figure 3A), while the other samples showed no significant differences (*p* > 0.05). By day 3, CON-DP-CAP recorded the highest PV (6.67 mg cumene hydroperoxide equivalent/kg sample), suggesting that CAP contributed to enhanced lipid oxidation. However, AGM-EP samples treated with CEC and CAP exhibited a lower PV compared to CON-DP-CAP (*p* < 0.05), indicating the radical-scavenging properties of CEC. As the storage duration increased, PV consistently rose across all samples (*p* < 0.05), likely due to the formation of hydroperoxide or peroxide compounds during the oxidation of lipids present in AGM [56]. Additionally, the abstraction of hydrogen from unsaturated fatty acids results in free radical formation. These highly reactive radicals readily interact with oxygen, resulting in the production of fatty acid hydroperoxides, which contribute to oxidative deterioration and impact meat quality [57]. By day 6, AGM-EP samples treated with CEC followed by CAP demonstrated a lower PV than the control samples (*p* < 0.05), and CEC-3-CAP showing the lowest PV. The presence of amino and hydroxyl groups in CEC enables it to scavenge free radicals, thereby mitigating oxidative damage and preserving meat quality [27]. Furthermore, the combined action of CEC and CAP might decrease the activity of microbial and endogenous enzymes (such as lipases and phospholipases) that promote lipid hydrolysis into glycerol and free fatty acids, thereby contributed to reducing lipid oxidation. By day 9, a similar pattern persisted, where the PV inversely correlated with CEC concentration. However, from day 12 until the end of storage, the PV of CEC-3-CAP was unchanged (*p* > 0.05).

TBARS levels followed a pattern similar to PV, with initial values ranging from 2.12 to 2.22 mg MDA/kg sample on day 0 (Figure 3B). Over time, TBARS content increased significantly, particularly in the CON-DP and CON-DP-CAP samples (*p* < 0.05), due to the accumulation of secondary oxidation compounds such as ketones and aldehydes [58]. By day 3, TBARS levels were lower as the CEC concentration increased, with CON-DP and CON-DP-CAP showing higher values (*p* < 0.05). As storage continued to day 6, CON-DP-CAP exhibited the highest TBARS content (*p* < 0.05), while other samples also showed notable increases. However, CEC-3-CAP had the lowest TBARS levels among all the samples (*p* < 0.05). This trend persisted through day 9, suggesting that CEC effectively inhibited secondary lipid oxidation. On days 12 and 15, TBARS levels in CEC-3-CAP remained below 4 mg MDA/kg sample. CEC likely mitigated lipid oxidation by scavenging free radicals produced by CAP treatment or limiting microbial growth, thereby reducing free fatty acid formation. Comparable effects were observed in our previous studies, where AGM or Asian seabass slices were exposed to CEC or CHOS treatment, where CEC or CHOS at different concentrations effectively mitigated the oxidative radicals produced by cold atmospheric plasma ([5] or [26], respectively). Additionally, the application of nanoliposomes incorporating a chitosan–EGCG conjugate helped suppress lipid oxidation in Asian seabass slices stored under refrigeration [59]. In a separate study, Khan et al. [60] reported that treating blue mussels (*Mytilus edulis*) with 0.01 M ascorbic acid during ice storage delayed lipid oxidation by five days, highlighting the potent antioxidant properties of ascorbic acid. These findings highlight the ability of CEC to suppress lipid oxidation in the AGM-EP, counteracting the effects of CAP treatment and microbial spoilage.

### 3.4. Changes in Fatty Acid Profiles

The fatty acid (FA) composition of the control samples (CON-DP and CON-DP-CAP) and CEC-3-CAP at day 0, along with CEC-3-CAP stored for 12 days at 4 °C, is given in Table 1. All the samples contained similar FAs on day 0, of which the pentadecanoic, palmitoleic, stearic, oleic, linolenic, linoleic, eicosapentaenoic (EPA), and docosahexaenoic (DHA) were the major FAs. On day 0, EPA and DHA varied from 12.02 to 12.42 mg/g and from 12.90 to 13.58 mg/g, respectively in all samples. Moreover, a lower content of saturated FAs (SFAs), monounsaturated FAs (MUFAs), and polyunsaturated FAs (PUFAs) was present in CON-DP-CAP (*p* < 0.05). A similar trend was noticed in our previous study, where AGM was depurated using CEC solutions at various levels and stored for 6 days [5]. Thus, it was presumed that CAP caused lipid oxidation to some extent. However, SFAs, MUFAs, and PUFAs were higher in CEC-3-CAP than CON-DP-CAP (*p* < 0.05), indicating that CEC could impede the adverse impact CAP. Likewise, Singh and Benjakul [26] demonstrated that CHOS treatment helped preserve the fatty acid profile of Asian seabass fillets during prolonged storage, minimizing compositional degradation over time. On day 12, CEC-3-CAP had a lower content of SFAs, MUFAs, and PUFAs than on day 0 (*p* < 0.05). It was found that linoleic acid was not detected on day 12. The results aligned with peroxide or hydroperoxide generation due to lipid oxidation in the AGM-EP when stored for a longer period, as indicated by higher PV and TBARS values (Figure 3A and Figure 3B, respectively). The loss of linoleic acid and other PUFAs significantly impacts the nutritional quality of the AGM, as these fatty acids are crucial for human health, particularly for their role in cardiovascular and cognitive functions. The reduction in PUFAs further affects the overall lipid profile, potentially diminishing the health benefits of consuming stored mussels.

Nevertheless, lipid oxidation could be conquered by CEC possessed with both antimicrobial and antioxidant activities, and it played a crucial role in the abatement of spoilage and lipid oxidation of the AGM-EP during refrigerated storage.

### 3.5. Changes in Textural Properties, Cooking Loss, and Sensory Analysis

The firmness and toughness of the AGM-EP when subjected to CEC treatment before CAP in comparison with CON-DP and CON-DP-CAP is given in Table 2. The firmness of CON-DP and CON-DP-CAP were not varied (*p* > 0.05), indicating that CAP had no adverse impact on firmness. However, CEC treatment in combination with CAP significantly reduces firmness as compared to other samples on day 0 (*p* < 0.05). Both the toughness and firmness of CEC-3-CAP were decreased during extended refrigerated storage (day 12) as compared to the sample at day 0 (*p* < 0.05). During extended storage, the action of the proteolytic enzymes caused the denaturation of myofibrillar proteins or the disruption of the connective tissues, potentially leading to structural modifications such as the collapse and separation of muscle fibers or connective tissue in the AGM-EP. Consequently, this caused a reduction in the firmness and toughness of CEC-3-CAP. Buatong et al. [5] also noticed reduction in the textural properties of the AGM during extended storage. Moreover, a loss in muscle structure might be mediated by proteolysis due to proteases released by spoilage bacteria during storage. This could impact on the overall palatability for consumers in a negative way, due to the formation of soft-textured meat. However, treated samples were more likely to preserve those textural changes as compared to those without any treatment.

Cooking influenced the weight of the meat, mainly due to the heat-induced denaturation of myofibrillar proteins, which led to a reduction in water-holding capacity. On day 0, the cooking loss ranged from 41.36% to 43.50% in all samples (*p* > 0.05; Table 2). Generally, cooking loss in the meat increased with the augmentation of storage time. This might be due to the heat denaturation of myofibrillar proteins, in which protein underwent aggregation as induced by heating, thus losing water-binding sites. In addition, hydrophobic residues might be more exposed, as induced by degradation. As a result, water-holding capacity became lower when heat was applied, and water in the meat was drastically liberated from the loose matrix with ease. Short-chain peptides generated via the degradation of the protein could not withstand the myofibrillar protein structure in meat effectively. The destruction of protein negatively affects the water-holding capacity of the mussel. However, cooking loss decreased to 31.52% when the AGM-EP was treated with CEC at 3% followed by CAP during refrigerated storage for 12 days (*p* < 0.05). In a previous study, Buatong et al. [5] also noticed a reduction in cooking loss from 43.71 % to 39.76%, when AGM treated with CEC (2%) was stored for 4 days. This could be associated with the loss of free water during storage (drip loss). Furthermore, CEC may enhance moisture retention, possibly by reinforcing protein integrity and water-holding capacity during storage. Singh and Benjakul [26] observed an enhanced water holding capacity when surimi gel was added with the chitooligosaccharide from the squid pen.

Likeness scores for appearance, color, odor, texture, taste, and overall, of all samples were in the ranges of 6.83–7.37, 6.45–7.71, 6.67–7.04, 6.54–7.04, and 6.96–7.25, respectively. On day 0, AGM-EP with CAP treatment without CEC (CON-DP-CAP) did not show a difference in the appearance, color, and texture likeness scores of cooked samples as compared to CON-DP (*p* > 0.05). Thus, CAP treatment did not negatively affect the likeness scores, confirming the acceptability of the AGM-EP. When Asian sea bass was treated with CAP, no difference in the likeness score of samples treated without and with CAP was noticed for all of the attributed sensory tests [26]. However, when AGM-EP was treated with CEC followed by CAP, taste, color, and appearance scores showed a significant decline (*p* < 0.05), though they remained above the acceptable limit. This decline may be attributed to the brownish color and astringent taste associated with CEC, which could have negatively impacted consumer perception. The adverse impact of CEC could be mitigated through encapsulation, which provides the sustained release of CEC while maintaining taste and color. Moreover, the overall acceptability, along with all sensorial attributes, of CEC-3-CAP dropped by day 12. It was slightly higher than the acceptable limit, which is 5.0. These decreases were linked to the accumulation of TVB-N, a byproduct of protein breakdown and lipid oxidation, both of which negatively affect the sensory characteristics—such as aroma, flavor, and palatability—of seafood during storage. Buatong et al. [5] also noticed the reduction in sensory score for AGM depurated using a 2% CEC solution during the refrigerated storage. Although sensory evaluation provided initial insights into consumer acceptance, further studies employing trained panels and advanced sensory profiling techniques are needed to more comprehensively evaluate the textural, flavor, and appearance attributes of CEC-treated mussels.

## 4. Conclusions

The combination of CEC at 3% (*w*/*w*) and CAP (CEC-3-CAP) effectively prolonged the shelf-life of the AGM-EP during refrigerated storage by reducing microbial proliferation and lipid oxidation. CEC abated adverse effects such as lipid oxidation induced by CAP due to the presence of amino and hydroxyl groups, which were able to scavenge free radicals that cause lipid oxidation. FAs, especially PUFAs, were retained in CEC-3-CAP after 12 days of storage. From a practical perspective, these findings suggest that incorporating CEC in combination with CAP could offer a natural preservation strategy for the AGM-EP, reducing reliance on synthetic antioxidants and antimicrobial agents. The ability of CEC-3-CAP to extend its microbial shelf-life by 12 days, while maintaining sensory properties, provides a viable approach for food producers to enhance products’ shelf-life. However, the likeness score of the treated samples was just higher than the acceptable limit after 12 days of storage, which requires further optimization or processing, such as the encapsulation of CEC to improve consumer perception. Overall, this study successfully demonstrates the potential of CEC as a natural additive for improving the storage stability of the AGM-EP by reducing lipid oxidation and microbial spoilage. However, further antimicrobial mechanisms of combined treatment should be determined for a better understanding of the preservative impact of CAP and CEC. Additionally, scaling up the process for industrial application and evaluating its cost-effectiveness will be crucial for widespread adoption in the food industry.

## Figures and Tables

**Figure 1 foods-14-01399-f001:**
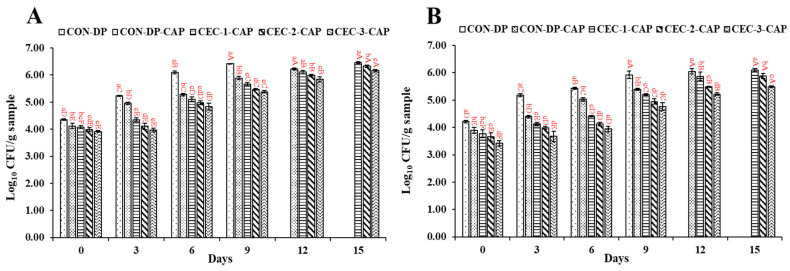
Changes in total viable bacteria (**A**), psychrophilic bacteria (**B**), *Vibrio* spp. (**C**), presumptive *Pseudomonas* (**D**), and H_2_S-producing bacteria (**E**) counts of the depurated Asian green mussel edible portion (AGM-EP) when treated with chitooligosaccharide–epigallocatechin gallate conjugate (CEC) at different concentrations in conjugation with cold atmospheric plasma (CAP) stored for 15 days at 4 °C. Bars represent the standard deviation (n = 3). Different lowercase letters on the bars within the same storage time indicate significant differences (*p* < 0.05). Different uppercase letters on the bars within the same sample indicate significant differences (*p* < 0.05). CON-DP: depurated AGM-EP; CON-DP-CAP: depurated AGM-EP treated with CAP in absence of CEC; CEC-1-CAP, CEC-2-CAP, and CEC-3-CAP: depurated AGM-EP treated with 1, 2, and 3% CEC combined with CAP, respectively.

**Figure 2 foods-14-01399-f002:**
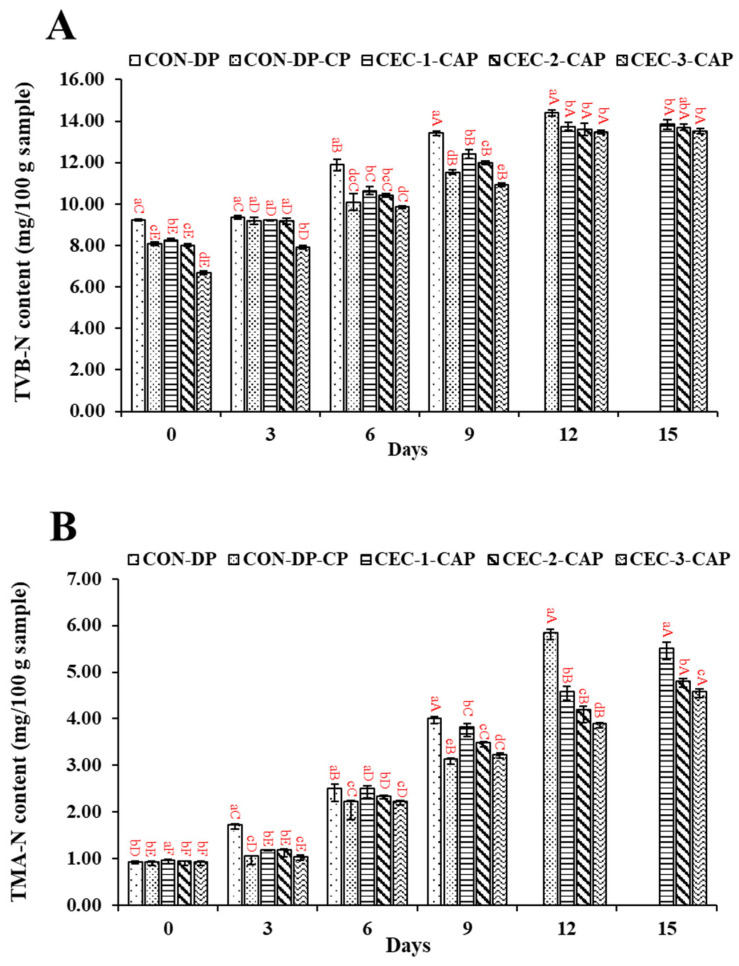
Changes in total volatile nitrogen base (TVB-N) (**A**) and trimethylamine (TMA-N) (**B**) of the Asian green mussel edible portion (AGM-EP) when treated with chitooligosaccharide–epigallocatechin gallate conjugate (CEC) at different concentrations in conjugation with cold atmospheric plasma (CAP) stored for 15 days at 4 °C. Bars represent the standard deviation (n = 3). Different lowercase letters on the bars within the same storage time indicate significant differences (*p* < 0.05). Different uppercase letters on the bars within the same sample indicate significant differences (*p* < 0.05). For captions, see Figure 1.

**Figure 3 foods-14-01399-f003:**
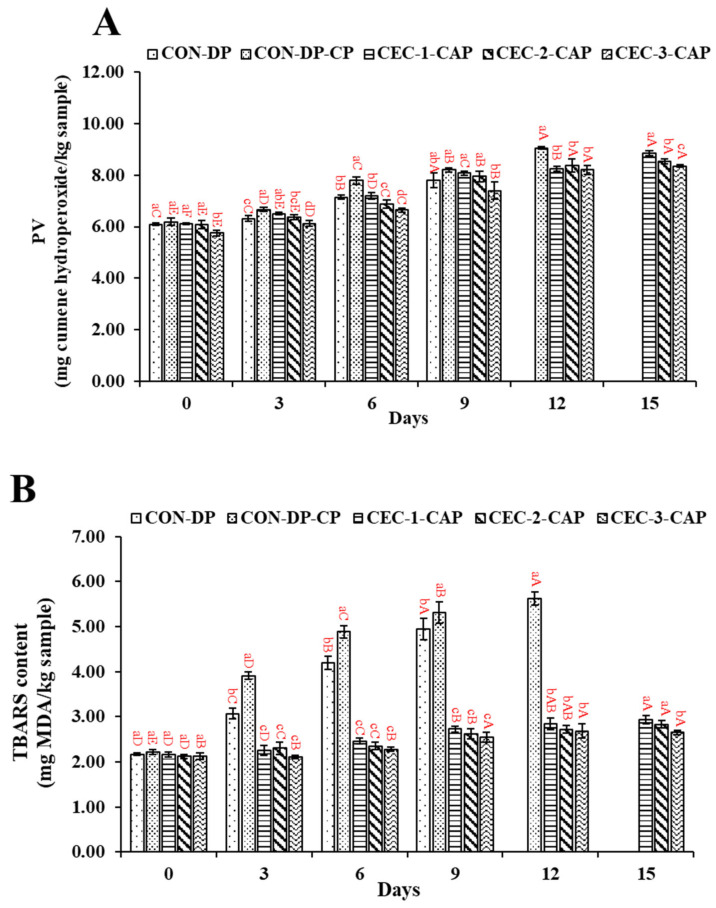
Changes in peroxide value (**A**) and thiobarbituric acid reactive substance (TBARS) content (**B**) of the depurated Asian green mussel edible portion (AGM-EP) when treated with chitooligosaccharide–epigallocatechin gallate conjugate (CEC) at different concentrations in conjugation with cold atmospheric plasma (CAP) stored for 15 days at 4 °C. Bars represent the standard deviation (n = 3). Different lowercase letters on the bars within the same storage time indicate significant differences (*p* < 0.05). Different uppercase letters on the bars within the same sample indicate significant differences (*p* < 0.05). For captions, see Figure 1.

**Table 1 foods-14-01399-t001:** Fatty acid composition of depurated Asian green mussel’s edible portion treated without and with CEC in combination with cold atmospheric plasma at days 0 and 12 while stored at 4 °C.

Fatty Acids (mg/g)	Day 0			Day 12
	CON-DP	CON-DP-CAP	CEC-3-CAP	CEC-3-CAP
C14:0 (Myristic Acid)	7.90 ± 1.05 ^b^	7.73 ± 0.09 ^bc^	10.62 ± 0.58 ^aA^	6.43 ± 0.07 ^B^
C14:1 (Myristoleic Acid)	1.14 ± 0.29 ^a^	0.91 ± 0.01 ^a^	1.08 ± 0.03 ^aA^	0.85 ± 0.16 ^B^
C15:1 (*cis*-10-Pentadecanoic Acid)	33.24 ± 2.15 ^b^	32.19 ± 0.17 ^a^	33.10 ± 1.00 ^bA^	32.61 ± 0.49 ^A^
C16:1 (Palmitoleic Acid)	1.09 ± 0.15 ^c^	0.94 ± 0.01 ^c^	1.13 ± 0.04 ^cA^	1.29 ± 0.14 ^A^
C17:0 (Heptadecanoic Acid)	2.66 ± 0.01 ^b^	1.57 ± 0.52 ^c^	2.06 ± 0.31 ^bcA^	ND ^B^
C17:1 (*cis*-10-Heptadecanoic Acid)	1.87 ± 0.20 ^c^	1.55 ± 0.08 ^c^	1.14 ± 0.09 ^dA^	1.19 ± 0.09 ^A^
C18:0 (Stearic Acid)	6.85 ± 0.95 ^ab^	6.30 ± 0.05 ^b^	7.82 ± 0.06 ^aA^	1.27 ± 0.21 ^B^
C18:1 (Oleic Acid)	2.93 ± 0.40 ^b^	2.96 ± 0.04 ^b^	4.60 ± 0.63 ^aA^	2.99 ± 0.74 ^B^
C18:2 (Linoleic Acid)	1.48 ± 0.20 ^a^	1.39 ± 0.02 ^a^	1.76 ± 0.29 ^aA^	ND ^B^
C18:3 (gamma-Linolenic Acid)	5.34 ± 0.75 ^a^	5.22 ± 0.07 ^a^	7.12 ± 0.68 ^aA^	4.61 ± 0.05 ^B^
C20:1 (cis-11-Eicosenoic Acid)	2.06 ± 0.30 ^b^	1.80 ± 0.03 ^b^	2.81 ± 0.52 ^aA^	1.94 ± 0.03 ^B^
C20:4 (*cis*-5,8,11,14-Eicosatetraenoic)	5.07 ± 0.75 ^ab^	4.67 ± 0.07 ^ab^	3.66 ± 0.03 ^cA^	4.41 ± 0.06 ^B^
C20:5 (*cis*-5,8,11,14,17-Eicosapentaenoic Acid)	12.42 ± 1.42 ^bc^	12.15 ± 0.17 ^ab^	12.02 ± 0.80 ^c^	12.07 ± 0.08
C22:6 (*cis*-4,7,10,13,16,19-Docosahexaenoic Acid)	13.58 ± 0.06 ^a^	12.08 ± 0.16 ^c^	12.90 ± 0.49 ^bA^	12.59 ± 0.17 ^A^
C23:0 (Tricosanoic)	2.37 ± 0.56 ^a^	1.54 ± 0.06 ^b^	2.02 ± 0.09 ^abA^	1.74 ± 0.06 ^B^
Saturated Fatty Acids	19.78 ± 1.46 ^b^	17.13 ± 0.31 ^c^	22.52 ± 1.04 ^aA^	9.44 ± 0.20 ^B^
Monounsaturated Fatty Acids	42.33 ± 1.80 ^b^	40.36 ± 0.17 ^c^	43.85 ± 0.02 ^aA^	40.87 ± 0.16 ^B^
Polyunsaturated Fatty Acids	37.90 ± 0.34 ^a^	35.51 ± 0.14 ^c^	37.45 ± 0.04 ^bA^	33.68 ± 0.37 ^B^

The results are presented as means ± SD (n = 3). Different lowercase letters in the same row within the same storage time indicate significant differences between different samples (*p* < 0.05). Different uppercase letters in the same row within the same sample indicate significant differences between different storage days (*p* < 0.05). Caption: see Figure 1.

**Table 2 foods-14-01399-t002:** Cooking loss, textural properties, and sensory analysis of cooked depurated Asian green mussel’s edible portion treated without and with CEC in combination with cold atmospheric plasma at days 0 and 12 while stored at 4 °C.

		Day 0			Day 12
		CON-DP	CON-DP-CAP	CEC-3-CAP	CEC-3-CAP
	Cooking loss (%)	42.28 ± 1.03 ^a^	43.50 ± 1.08 ^a^	41.36 ± 0.96 ^a^	31.52 ± 2.71 ^b^
Textural properties	Firmness (g)	1881.67 ± 16.64 ^a^	1898.97 ± 6.65 ^a^	1744.52 ± 17.69 ^b^	622.19 ± 56.21 ^c^
	Toughness (g)	9149.20 ± 43.69 ^a^	8404.34 ± 92.84 ^b^	7819.12 ± 79.36 ^c^	818.24 ± 23.06 ^d^
Sensory analysis	Appearance	7.37 ± 0.71 ^a^	7.33 ± 0.70 ^a^	6.83 ± 0.91 ^aA^	5.50 ± 1.21 ^B^
	Color	7.17 ± 1.05 ^a^	7.08 ± 0.83 ^ab^	6.45 ± 0.97 ^bA^	5.75 ± 0.90 ^B^
	Odor	7.25 ± 0.99 ^a^	6.91 ± 0.82 ^ab^	6.79 ± 0.83 ^bA^	5.37 ± 1.13 ^B^
	Texture	7.04 ± 0.91 ^a^	6.92 ± 0.83 ^a^	6.67 ± 1.09 ^aA^	5.08 ± 0.97 ^B^
	Taste	7.04 ± 0.81 ^a^	6.87 ± 0.61 ^ab^	6.54 ± 0.78 ^bA^	5.00 ± 0.88 ^B^
	Overall likeness	7.25 ± 0.53 ^a^	7.17 ± 0.70 ^a^	6.96 ± 0.81 ^aA^	5.21 ± 0.77 ^B^

The results are presented as means ± SD (n = 3 and n = 50 for sensory analysis). Different lowercase letters in the same row denote significant differences (*p* < 0.05). Different uppercase letters in the same row denote significant differences between the same samples on different storage days. Caption: see Figure 1.

## Data Availability

The original contributions presented in this study are included in the article, and further inquiries can be directed to the corresponding author.

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
