# Peer review of "Combined Effect of Cold Atmospheric Plasma and Chitooligosaccharide–EGCG Conjugate on Quality and Shelf-Life of Depurated Asian Green Mussel"

_foods, 2025, doi:10.3390/foods14081399_

Round 1
Reviewer 1 Report
Comments and Suggestions for Authors
After conducting a critical and constructive evaluation of the manuscript, it is clear that the topic is both relevant and timely. The literature highlights the benefits of using high-voltage cold plasma (HVCP) and chitooligosaccharides (CHOS) separately to improve the quality of various seafood products. Therefore, one would expect that combining both treatments would yield a similar or even synergistic effect that could further enhance the quality and shelf-life of seafood products. It is essential that the authors clearly justify in the manuscript the rationale for combining these treatments, especially if the individual effects are already known to be beneficial. Relevant studies that support this point include:
Buatong, J., Bahem, N., Benjakul, S., Patil, U., & Singh, A. (2024). Depuration of Asian Green Mussels Using Chitooligosaccharide-Epigallocatechin Gallate Conjugate: Shelf-Life Extension, Microbial Diversity, and Quality Changes during Refrigerated Storage. Foods, 13(19), 3104.
Shiekh, K. A., & Benjakul, S. (2020). Effect of high voltage cold atmospheric plasma processing on the quality and shelf-life of Pacific white shrimp treated with Chamuang leaf extract. Innovative Food Science & Emerging Technologies, 64, 102435.
Regarding the manuscript title, it is too long. I suggest shortening it to improve clarity and impact.
In the abstract, it would be highly beneficial to include numerical data from the results, such as the extent to which shelf-life was extended. This would enhance comprehension and allow a better assessment of the study’s outcomes.
On line 16, “volatile bases” should be corrected to “total volatile bases”. Additionally, clarification is needed: does TVB-N only include TMA-N and TVB-N, or are other compounds involved?
In the Introduction, the authors could improve the section by including the market price and consumption rate of Perna viridis, as well as a brief explanation of why this species is highly perishable. The section is composed of two overly long paragraphs—these should be shortened for better readability. Although the effects of CAP and CHOS are presented individually, the authors do not explain the rationale for studying their combination, which should be addressed explicitly.
In the Materials and Methods, on line 107, the authors should specify how many mussels were used and provide information about the maturity stage of the animals. Section 2.3 is unclear and needs significant improvement; each treatment should be justified clearly. On line 121, the authors should explain why 16 kV was chosen for the plasma treatment. In sections 2.4.4, 2.4.5, and 2.4.7, the authors should provide references for each analytical method. In section 2.5, they should indicate which software was used for the statistical analysis.
In the Results, some paragraphs are too long and should be significantly shortened. The results are also written mostly in a descriptive manner; many statements are limited to noting that values increased or decreased, without explaining the underlying biological, chemical, or technological reasons. I strongly recommend a more in-depth analysis of the data, supported by a comprehensive discussion and comparisons with findings reported in previous studies. It would also be very valuable to include an integrated analysis of the results, such as a correlation matrix and principal component analysis (PCA).
Based on the points above, I suggest that the authors revise the abstract and conclusions, focusing on the main findings and their scientific and practical relevance.
Comments on the Quality of English Language
the manuscript has a good level of English
Author Response
Reviewer 1
Thank you for the constructive feedback. We appreciate the recognition of the relevance and timeliness of our topic. All the queries have been responded, necessary changes were done as suggested by reviewer and highlighted in blue color.
After conducting a critical and constructive evaluation of the manuscript, it is clear that the topic is both relevant and timely. The literature highlights the benefits of using high-voltage cold plasma (HVCP) and chitooligosaccharides (CHOS) separately to improve the quality of various seafood products. Therefore, one would expect that combining both treatments would yield a similar or even synergistic effect that could further enhance the quality and shelf-life of seafood products. It is essential that the authors clearly justify in the manuscript the rationale for combining these treatments, especially if the individual effects are already known to be beneficial. Relevant studies that support this point include:
Buatong, J., Bahem, N., Benjakul, S., Patil, U., & Singh, A. (2024). Depuration of Asian Green Mussels Using Chitooligosaccharide-Epigallocatechin Gallate Conjugate: Shelf-Life Extension, Microbial Diversity, and Quality Changes during Refrigerated Storage. Foods, 13(19), 3104.
Shiekh, K. A., & Benjakul, S. (2020). Effect of high voltage cold atmospheric plasma processing on the quality and shelf-life of Pacific white shrimp treated with Chamuang leaf extract. Innovative Food Science & Emerging Technologies, 64, 102435.
******Thank you for your comment. Although the justification for combining these two treatments already in the text, it has been improved for better understanding. Please see line 85-92 and 104-108.
Regarding the manuscript title, it is too long. I suggest shortening it to improve clarity and impact.
******The title has been edited. Please see the amended title.
In the abstract, it would be highly beneficial to include numerical data from the results, such as the extent to which shelf-life was extended. This would enhance comprehension and allow a better assessment of the study’s outcomes.
******The extent of shelf-life extended has been added to the abstract.
On line 16, “volatile bases” should be corrected to “total volatile bases”. Additionally, clarification is needed: does TVB-N only include TMA-N and TVB-N, or are other compounds involved?
******Thank you for your comment. The text has been updated as suggested by the reviewer. Please see line 16. In this study, total volatile bases were analyzed by measuring TVB-N (total volatile basic-N) and TMA-N (trimethylamine-N), while other compounds such as DMA-N, MMA-N, and ammonia were not included.
In the Introduction, the authors could improve the section by including the market price and consumption rate of Perna viridis, as well as a brief explanation of why this species is highly perishable. The section is composed of two overly long paragraphs—these should be shortened for better readability. Although the effects of CAP and CHOS are presented individually, the authors do not explain the rationale for studying their combination, which should be addressed explicitly.
******Thank you for your comment. Market value of Perna viridis is already provided in the text. Please see line 44. However, the data on consumption rate is not available, therefore, we cannot provide it. Apologies for this.
******The introduction section is mainly divided into two sections. In the first section, rationale behind selecting Perna viridis and challenges associated with it discussed while in second section potential of non-thermal processing and natural preservative such as chitooligosaccharide is given. Both sections provide comprehensive understanding of the context.
******The rationale behind studying combination is based on the mitigation of adverse impact such as lipid oxidation caused by cold atmospheric plasma by adding natural preservative, which can combat with antimicrobial resistance. The information was provided in the text. Please see line 85-92 and 104-108. Moreover, the rationale has been discussed further for better understanding. Please see line 111-116.
In the Materials and Methods, on line 107, the authors should specify how many mussels were used and provide information about the maturity stage of the animals. Section 2.3 is unclear and needs significant improvement; each treatment should be justified clearly. On line 121, the authors should explain why 16 kV was chosen for the plasma treatment. In sections 2.4.4, 2.4.5, and 2.4.7, the authors should provide references for each analytical method. In section 2.5, they should indicate which software was used for the statistical analysis.
******Thank you for your comment. The number of mussels and maturity stage has been added to the text. Please see line 114.
******The section 2.3 has been improved and provided additional details as suggested by the other reviewers.
******Generally, dielectric barrier discharge plasma requires high voltage (typically 5-30 kV) to ionize gases like air, argon, or nitrogen. In this study, the unit is equipped with high voltage power supply, which generates 16 kV RMS corresponding to 22.6 kV peak in a sinusoidal waveform. This voltage is sufficient to create plasma by breaking down the gas between electrodes.
******The references have been added to text. Please see the mentioned sections.
******The name of software used for statistical analysis has been added to the text.
In the Results, some paragraphs are too long and should be significantly shortened. The results are also written mostly in a descriptive manner; many statements are limited to noting that values increased or decreased, without explaining the underlying biological, chemical, or technological reasons. I strongly recommend a more in-depth analysis of the data, supported by a comprehensive discussion and comparisons with findings reported in previous studies. It would also be very valuable to include an integrated analysis of the results, such as a correlation matrix and principal component analysis (PCA).
******Thank you for your comment. Authors would be appreciable if reviewer precisely point out paragraphs that needed to shorten. Moreover, authors carefully read the manuscript again and all the results discussed with possible reasons and supported by relevant references. The possible mechanism of microbial inhibition is explained in line 204-206, 209-216, 281-284, 292-295. Similarly other results were supported with appropriate discussion and findings reported in previous studies.
******Integrating correlation matrix and PCA would be a great approach for more comprehensive analysis of the data. Authors acknowledge reviewer’s suggestion and positively incorporate in their future studies, however, in this study authors postulated general correlation between microbial and quality parameters, in which increase in microbial load leads to detrimental effect on the quality of AGM-EP such as higher volatile bases, lipid oxidation, etc. and reduced shelf-life.
Based on the points above, I suggest that the authors revise the abstract and conclusions, focusing on the main findings and their scientific and practical relevance.
******The abstract and conclusion has been revised to provide main findings and their scientific and practical relevance. Please see the yellow highlighted text in conclusion section.
Reviewer 2 Report
Comments and Suggestions for Authors
Line 107: indicate please, the sample size.
Line 107: where the AGM sampled in the same day or in different days?
Line 189-192: I suggest including the characteristics of the column used and all the GC settings; e.g. temperature, carrying gas etc.
Line 202: authors used a Completely randomized design for all experiments and analyses an all figures and tables shows two comparisons; one for the treatments and one for the days. Authors need to use a randomized completely block design if they want to compare treatments on each storage day. However, a factorial design describes properly the experiment because the interactions between treatments and storage days can be assessed.
Line 203: sensory evaluation variables do not present normal distribution, therefore a randomized completely block design is not suitable for this statistical analysis. I recommend evaluating sensory evaluation variables with a non-parametric test like Friedman test an post hoc test Wilcoxon-Bonferroni or Conover-Iman.
Line 234: Figure 1 typing error … “different concenTable 15. Days”
Line 236: Figure 1 typing error… “letters on Table 0.”
Figures 1 to 4: explain why, there are four and three bars at 12 and 15 days of storage; respectively, rather than five bars.
Author Response
Reviewer 2
Thank you for your invaluable comments for improving manuscript. All the queries have been responded in the text and highlighted in pink color.
Line 107: indicate please, the sample size.
******The sample size (n=100) has been added to the text. Please see line 125.
Line 107: where the AGM sampled in the same day or in different days?
******The sampling was done on the same day. Please see line 126.
Line 189-192: I suggest including the characteristics of the column used and all the GC settings; e.g. temperature, carrying gas etc.
******Thank you for the comment. We have provided those details. Please see line 182-186.
Line 202: authors used a Completely randomized design for all experiments and analyses an all figures and tables shows two comparisons; one for the treatments and one for the days. Authors need to use a randomized completely block design if they want to compare treatments on each storage day. However, a factorial design describes properly the experiment because the interactions between treatments and storage days can be assessed.
****** Thank you for your comment. We used a CRD because it was simpler and allowed us to look at the overall treatment effects. However, we understand that RCBD would be a better choice if we want to compare treatments on each storage day. This would be accounted for the future studies. Thank you for your suggestion. Moreover, as you correctly pointed out, the factorial design used in this study does indeed allow for the assessment of interactions between treatments and storage days. We believe that the factorial design provides a clear understanding of how the treatments perform over time and how the two factors (treatment and storage day) influence the outcomes.
Line 203: sensory evaluation variables do not present normal distribution, therefore a randomized completely block design is not suitable for this statistical analysis. I recommend evaluating sensory evaluation variables with a non-parametric test like Friedman test an post hoc test Wilcoxon-Bonferroni or Conover-Iman.
******Thank you for your comment. Authors agreed that sensory evaluation data often does not follow a normal distribution, a randomized complete block design (RCBD), which typically assumes normality, may not be the best choice. Therefore, authors re-analyze the sensory evaluation data using non-parametric test like Friedman test and post hoc Wilcoxon-Bonferroni.
Line 234: Figure 1 typing error … “different concenTable 15. Days”
******Typing error has been resolved.
Line 236: Figure 1 typing error… “letters on Table 0.”
******Typing error has been resolved.
Figures 1 to 4: explain why, there are four and three bars at 12 and 15 days of storage; respectively, rather than five bars.
******Thank you for your comment. Authors selected samples those possessed total viable bacteria count less than 6 log CFU/g sample on preceding sampling day for analysis on day 12 and 15.

Reviewer 3 Report
Comments and Suggestions for Authors
This study investigated the combined effects of chitooligosaccharide-epigallocatechin gallate conjugate (CEC) and cold atmospheric plasma (CAP) on the physicochemical properties, microbiological quality, and shelf-life of depurated Asian green mussels during refrigerated storage. The research is innovative and the topic is highly significant, particularly due to the pressing need for effective preservation methods in the seafood industry. Detailed comments and requests for corrections are listed below:
Major comments and required corrections:
Introduction:
Provide more context or justification for selecting chitooligosaccharide-epigallocatechin gallate conjugate specifically over other potential antimicrobial and antioxidant agents (Lines 89-95).
Materials and Methods:
Specify clearly the rationale for using a combination of argon and oxygen gas in CAP treatment. Why was this specific ratio (90% Ar:10% O2) chosen? (Lines 118-120).
Provide additional details on the storage conditions (humidity, packaging specifications, etc.) during refrigerated storage (Lines 124-126).
Results and Discussion:
Address why microbial counts initially decreased on day 3, specifically Vibrio spp., and then provide more discussion on microbial dynamics throughout storage (Lines 263-266).
Explain the significant decrease in fatty acid content after 12 days, especially linoleic acid disappearance, and discuss its implications on the overall nutritional quality of the mussels (Lines 434-436).
Provide more insights into the reasons for the observed reduction in firmness and toughness and how these changes might practically affect consumer acceptance (Lines 447-453).
Explain more thoroughly why sensory attributes such as color, appearance, and taste scores decreased significantly when CEC was used, and consider discussing potential strategies to mitigate these sensory limitations (Lines 477-486).
Figures:
Figures 1 and 2 appear repetitive regarding data representation. Consider consolidating these results either in tables or figures for improved clarity and readability.
Conclusion:
Offer clear, practical suggestions for further studies, specifically addressing how future research might optimize the CEC and CAP process to retain better sensory and textural properties of the mussels (Lines 491-502).
The similarity index detected by iThenticate is currently very high (46%) and must be significantly reduced.
Author Response
Reviewer 3
This study investigated the combined effects of chitooligosaccharide-epigallocatechin gallate conjugate (CEC) and cold atmospheric plasma (CAP) on the physicochemical properties, microbiological quality, and shelf-life of depurated Asian green mussels during refrigerated storage. The research is innovative and the topic is highly significant, particularly due to the pressing need for effective preservation methods in the seafood industry. Detailed comments and requests for corrections are listed below:
Major comments and required corrections:
******Thank you for the insightful comments and feedback, which can enhance the clarity of our manuscript. All queries have been answered. The corrections have been done, and the additional information have been provided as highlighted in green color.
Introduction:
Provide more context or justification for selecting chitooligosaccharide-epigallocatechin gallate conjugate specifically over other potential antimicrobial and antioxidant agents (Lines 89-95).
******Thank you for your comment. Chitooligosaccharide-epigallocatechin gallate conjugate has been explored for its health related activities in both in vitro and in vivo models, which have been added to the text. Please see line 107-109.
Materials and Methods:
Specify clearly the rationale for using a combination of argon and oxygen gas in CAP treatment. Why was this specific ratio (90% Ar:10% O2) chosen? (Lines 118-120).
******The selection of gas ratio (90% Ar:10% O2) is based on previous studies, which showed higher antimicrobial activities with lower lipid and protein oxidations. The text has been updated, please see line 132-133.
- Olatunde, O. O., Benjakul, S., & Vongkamjan, K. (2019). High voltage cold atmospheric plasma: Antibacterial properties and its effect on quality of Asian sea bass slices. Innovative Food Science & Emerging Technologies, 52, 305-312.
- Singh, A., & Benjakul, S. (2020). The combined effect of squid pen chitooligosaccharides and high voltage cold atmospheric plasma on the shelf-life extension of Asian sea bass slices stored at 4 °C. Innovative Food Science & Emerging Technologies, 64, 102339.
- Olatunde, O. O., Benjakul, S., & Vongkamjan, K. (2019). Combined effects of high voltage cold atmospheric plasma and antioxidants on the qualities and shelf-life of Asian sea bass slices. Innovative food science & emerging technologies, 54, 113-122.
Provide additional details on the storage conditions (humidity, packaging specifications, etc.) during refrigerated storage (Lines 124-126).
******The additional details on the storage conditions and packaging specifications have been to the text. Please see lines 141 and 151.
Results and Discussion:
Address why microbial counts initially decreased on day 3, specifically Vibrio spp., and then provide more discussion on microbial dynamics throughout storage (Lines 263-266).
******Thank you for your comment. Most of the counts are explained with the reason for lower microbial count on day 3 (line 212-214 and 217-224). However, considering reviewers comment, the discussion has been rephrased, in which reason behind a decrease in microbial count on day 3 for Vibrio spp. has been mentioned. Please see line 263-270.
Explain the significant decrease in fatty acid content after 12 days, especially linoleic acid disappearance, and discuss its implications on the overall nutritional quality of the mussels (Lines 434-436).
******The relevant discussion was already given in line 434-436. Moreover, implications of decrease in PUFA on the overall nutritional quality of AGM have been added to the text. Please see line 436-440.
Provide more insights into the reasons for the observed reduction in firmness and toughness and how these changes might practically affect consumer acceptance (Lines 447-453).
******Thank you for your comment. The discussion was given in the text. Please see line 455-459 and 460-463.
Explain more thoroughly why sensory attributes such as color, appearance, and taste scores decreased significantly when CEC was used, and consider discussing potential strategies to mitigate these sensory limitations (Lines 477-486).
******The reason behind decrease in sensory attributes upon adding CEC to the sample was given in line 490-491. Moreover, a possible strategy to overcome the limitation of CEC has been to the text. Please see line 492-493.
Figures:
Figures 1 and 2 appear repetitive regarding data representation. Consider consolidating these results either in tables or figures for improved clarity and readability.
****** Thank you for your feedback. Figure 1 and 2 have been merged to improve clarity and readability.
Conclusion:
Offer clear, practical suggestions for further studies, specifically addressing how future research might optimize the CEC and CAP process to retain better sensory and textural properties of the mussels (Lines 491-502).
Authors revised the Conclusion with emphasizing the practical implications of current study, demonstrating how the data can be applied in real-world scenarios for better understanding.

Reviewer 4 Report
Comments and Suggestions for Authors
There are many concerns about the MS. Here, the main ones. In any research concerning a living organism, the identification of the species should be determined by an expert in the field and a voucher deposited for further control. Who is the expert in this case? The seller? Temperatures should be rewritten, like 4°C as 4 °C. There are some typo errors in the MS. The same argument for the strains of microorganisms. The sensory evaluation has evident limits. The Introduction is too long and this way the relevance and topic of the research are not clear to the reader. I suggest to focus on the organisms in the research. Abbreviations of several terms are used. Therefore, a list of them should be reported, to help the reader in following the MS, which is quite complex and this way the abbreviations should be used in all parts of the text, avoiding several repetitions, like in figures. It was difficult for me to fully understand the meaning of some parts of the MS. Like in the sentences This increase is likely due to lipid oxidation 387 within AGM, leading to the formation of hydroperoxide or peroxide compounds [58]. Ad- 388 ditionally, the abstraction of hydrogen from unsaturated fatty acids results in free radical 389 formation, which, upon reacting with oxygen, produces fatty acid hydroperoxides [59]. 390 By day 6, AGM-EP samples treated with CEC followed by CAP demonstrated lower PV 391 than the control samples (p<0.05), with CEC-3-CAP showing the lowest PV among the 392 AGM-EP samples treated with CEC and HVCAP. The presence of amino and hydroxyl 393 groups in CEC enables it to scavenge free radicals and donate hydrogen atoms to them 394 [29]. Furthermore, the combined action of CEC and CAP on microbial and endogenous 395 enzymes (such as lipases and phospholipases) likely contributed to reducing lipid oxida- 396 tion. By day 9, a similar pattern persisted, where PV values inversely correlated with CEC 397 concentration. However, from day 12 until the end of storage, PV values for CEC-3-CAP 398 showed a steady increase (p>0.05). The mechanism of oxidation of fatty acids is well known, but it is not the same for any kind of lipids. However, the main concern is the about the significance of these parts, i.e. if these are general interpretations of the authors or they are supported by data from the research. In fact, these aspects are reported in the Conclusion. In the Conclusions, the authors should be able to explain to the reader how in practice the data reported should be evidenced, since probably the reader or a producer could be interested in the real possible impact of the research. The tendency to report the results of the research is a typical attitude and the consequence is that most of the research remains at the level of apeace of paper. In Table 1, cis should be written in italic.
Author Response
Reviewer 4
There are many concerns about the MS.
******Authors would like to express our gratitude for the insightful comments and suggestions. All queries have been responded, and the corrections have been highlighted in yellow color.
Here, the main ones. In any research concerning living organisms, the identification of the species should be determined by an expert in the field and a voucher deposited for further control. Who is the expert in this case? The seller?
******The sample selection was done by the expert guidelines as well as the experienced seller. Furthermore, authors have identified the Asian green mussel morphologically by authors as described by Carpenter and Niem (1998). The relevant information has been added to the text for better understanding. Please see line 116-119.
- E. Carpenter, V. H. Niemeditors, 19990104207, English, Book, Italy, 9789251040515, Rome, The living marine resources of the Western Central Pacific. Volume 1. Seaweeds, corals, bivalves and gastropods., (xiv + 686 pp.), Food and Agriculture Organization of the United Nations, The living marine resources of the Western Central Pacific. Volume 1. Seaweeds, corals, bivalves and gastropods., (1998)
Temperatures should be rewritten, like 4°C as 4 °C.
******Temperature has been rewritten as 4°C instead of 4 °C.
There are some typo errors in the MS.
******The manuscript has been read thoroughly to rectify typo errors.
The same argument for the strains of microorganisms.
******All typo errors related to microorganisms have been rectified.
The sensory evaluation has evident limits.
******Authors acknowledge that sensory evaluation has certain limitations such as subjectivity, panelist variability including personal biasness, individual preferences, cultural backgrounds, etc. However, since the analysis was conducted by the untrained panelists familiar with seafood flavors, the findings still offer valuable insights into consumer perception and product quality, such as the limitation of CEC on Asian green taste.
******Authors have updated the text to justify this content. Please see line 497-501.
The Introduction is too long and this way the relevance and topic of the research are not clear to the reader. I suggest to focus on the organisms in the research.
******Authors thoroughly read the introduction section of the manuscript, and their intention was to provide a comprehensive background to establish the context of the research. However, authors agreed with the reviewer comment, therefore Introduction has been revised concisely while ensuring that the key points are effectively communicated.
Abbreviations of several terms are used. Therefore, a list of them should be reported, to help the reader in following the MS, which is quite complex and this way the abbreviations should be used in all parts of the text, avoiding several repetitions, like in figures.
******Abbreviations list has been added to text. Additionally, abbreviations have been rechecked throughout the manuscripts including figures and tables to ensure uniformity. Please see line 35-40.
It was difficult for me to fully understand the meaning of some parts of the MS. Like in the sentences This increase is likely due to lipid oxidation 387 within AGM, leading to the formation of hydroperoxide or peroxide compounds [58]. Ad- 388 ditionally, the abstraction of hydrogen from unsaturated fatty acids results in free radical 389 formation, which, upon reacting with oxygen, produces fatty acid hydroperoxides [59]. 390 By day 6, AGM-EP samples treated with CEC followed by CAP demonstrated lower PV 391 than the control samples (p<0.05), with CEC-3-CAP showing the lowest PV among the 392 AGM-EP samples treated with CEC and HVCAP. The presence of amino and hydroxyl 393 groups in CEC enables it to scavenge free radicals and donate hydrogen atoms to them 394 [29]. Furthermore, the combined action of CEC and CAP on microbial and endogenous 395 enzymes (such as lipases and phospholipases) likely contributed to reducing lipid oxida- 396 tion. By day 9, a similar pattern persisted, where PV values inversely correlated with CEC 397 concentration. However, from day 12 until the end of storage, PV values for CEC-3-CAP 398 showed a steady increase (p>0.05). The mechanism of oxidation of fatty acids is well known, but it is not the same for any kind of lipids. However, the main concern is the about the significance of these parts, i.e. if these are general interpretations of the authors or they are supported by data from the research. In fact, these aspects are reported in the Conclusion.
******The authors carefully reviewed and edited the highlighted part of the manuscript for better understanding. Authors agreed that oxidation mechanisms are dependent on the type of fatty acid but here the case of food lipids, especially fatty acids. The possible mechanism of fatty acids oxidation was explained in the text. Furthermore, the statements made in the discussion and conclusion are not general assumptions but are supported by our experimental data such as PUFA, PV and TBARS., which support our objective. For further explanation based on individual types of lipid, a separate study is required in depth. Please see line 380-394. Moreover, interpretations have been included in the conclusion section for better understanding.
In the Conclusions, the authors should be able to explain to the reader how in practice the data reported should be evidenced, since probably the reader or a producer could be interested in the real possible impact of the research. The tendency to report the results of the research is a typical attitude and the consequence is that most of the research remains at the level of apeace of paper.
******Thank you for your insightful comment. Authors appreciated your suggestion to enhance the Conclusion section by providing a clearer explanation of the practical implications of our findings. Authors revised the Conclusion with emphasizing the practical implications of current study, demonstrating how the data can be applied in real-world scenarios for better understanding. Please see line 508-514 and 519-520.
In Table 1, cis should be written in italic.
******‘cis’ has been italicized in Table 1.

Round 2
Reviewer 1 Report
Comments and Suggestions for Authors
1.- The study was conducted with n = 3. Statistically, is this sufficient to ensure the reliability of the data?
2.- In the results section, it is indicated that p > 0.05 is not significant and p < 0.05 is significant. If p = 0.05, is it considered significant or not? Please clarify and correct this throughout the results.
3.- Please divide the excessively long paragraphs between lines 43–70, 71–117, 204–224, 261–283, 315–341, and 372–394. Long paragraphs can be tedious to read.
4.- It is unclear how the authors concluded that the shelf-life of depurated AGM-EP is at least 9 days compared to the control (3 days). Could the authors clarify this point in the manuscript? Total volatile base nitrogen (TVB-N) values at the end of storage remain below the maximum permissible limit of 30 mg.
5.- Lines 52–54: The authors state that the quality deterioration of seafood products is due to high moisture content. I suggest revising this point and providing a more comprehensive explanation, as it is not solely related to moisture content.
6.- It would be valuable for the authors to include the chemical composition analysis of AGM in their study, as this would influence the choice of analytical methods. How do the authors justify its omission?
7.- Please provide more detailed descriptions of the methodologies mentioned in section 2.4.2.
8.- Line 193: On what basis did the authors determine that a sensory score below 5 on the evaluation scale indicates an unacceptable product? Please explain.
9.- Lines 236–249: Please discuss the reasons for your findings and compare them with those of other studies.
10.-Lines 270–271: Please include a reference.
11- Lines 280–283: Please include references.
12.Lines 285–287: Please include references.
13.-Lines 297–300: Please include a comparison with other studies and corresponding references.
14.- Lines 305–307: Please add a reference.
15.- Line 313: Please include a comparison with findings from other studies.
16.-Lines 398–408: Please compare your results with those in the existing literature.
17.- Lines 423–440: Could the authors add references and compare their results with those from other studies?
18.- Lines 446–463: Please include references and comparative analysis with prior studies.
19.- Lines 464–478: Please add references and comparisons with other studies.
20.- Lines 483–501: Please include references and compare these results with those reported in the literature.
21.-I would like to reiterate that performing a principal component analysis (PCA) would greatly enhance the quality of the manuscript.
22.- It is necessary to reduce the manuscript's similarity index, which currently stands at 33%.
Comments on the Quality of English LanguageThe English language is good.
Author Response
Response to reviewer 1
1.- The study was conducted with n = 3. Statistically, is this sufficient to ensure the reliability of the data?
*****We performed experiments in triplicate (n=3), which is a commonly opted methodology in the food shelf-life studies to ensure reliability while managing variability. Many similar studies report results based on triplicate analyses and authors considered this sufficient to elucidate the variability among treatments. We have clarified in the Methods that “all analyses were performed in triplicate” to emphasize the statistical robustness. Increasing the number of replicates can improve power, but even with n=3 we observed consistent, significant differences (p≤0.05) between treatments, indicating that our sample size was adequate to support the conclusions. This approach is in line with standard experimental designs in our field and is now explicitly stated in the revised manuscript.
2.- In the results section, it is indicated that p > 0.05 is not significant and p < 0.05 is significant. If p = 0.05, is it considered significant or not? Please clarify and correct this throughout the results.
The significance was set at p ≤ 0.05. Please see line 206.
3.- Please divide the excessively long paragraphs between lines 43–70, 71–117, 204–224, 261–283, 315–341, and 372–394. Long paragraphs can be tedious to read.
Thank you for your comment. As suggested by the reviewer, the paragraphs have been reorganized for better clarity and readability. Specifically, the section from lines 71–117 has been divided into three self-contained paragraphs: L75–97, L98–114, and L115–122. Each of these addresses a distinct and fully described topic. L43-74 includes basic introduction about AGM, L209–229 discusses the total viable bacteria count, L266–288 focuses on changes in Vibrio counts, L319–345 covers variations in total volatile bases (TVB), and L376–399 presents changes in peroxide value (PV). All of these paragraphs are approximately 20 lines in length and are structured to ensure that each discusses a complete and self-contained topic. Therefore, it is inappropriate for splitting aforementioned paragraphs.
4.- It is unclear how the authors concluded that the shelf-life of depurated AGM-EP is at least 9 days compared to the control (3 days). Could the authors clarify this point in the manuscript? Total volatile base nitrogen (TVB-N) values at the end of storage remain below the maximum permissible limit of 30 mg.
*****Thank you for your comment. Shelf-life in our study was primarily determined based on microbiological and sensory criteria. Specifically, we used the widely accepted threshold of ~6 log CFU/g for total viable counts (TVC) as the upper microbial limit for acceptability in seafood, in accordance with the guidelines of the International Commission on Microbiological Specifications for Foods (ICMSF). In our data, control sample approached this microbial threshold by day 3, while the CEC-3-CAP treated mussels remained well below this limit till day 12 of the storage period. In parallel, sensory evaluation—particularly hedonic scores for taste and odor—was also used to determine acceptability, with a score above 5 indicating acceptable quality. By day 12, the sensory scores for treated samples remained just above this threshold, though indicating reduced consumer appeal. Based on these combined criteria-microbial safety and sensory acceptability—we concluded that the shelf-life of the depurated AGM-EP is at least 9 days, as quality remained within acceptable limits through that period.
*****It is true that TVB-N were below the acceptable limits, which could be influence by several factors. The CEC coating likely inhibited the activity of spoilage bacteria responsible for producing volatile nitrogenous compounds, such as ammonia, dimethylamine, and trimethylamine, which constitute TVB-N. This inhibition could occur through several mechanisms:​
- Selective Antimicrobial Action: CEC may have selectively suppressed the growth or metabolic activity of specific spoilage organisms (SSOs) that are primarily responsible for the production of volatile amines, without significantly affecting the total viable count (TVC).​
- Enzymatic Activity Inhibition: The CEC and CAP are known to inhibit the enzymes, which could have impacted on the endogenous enzymatic activities, such as deaminases and decarboxylases, that contribute to the breakdown of proteins into volatile nitrogenous compounds.​
- Alteration of Microbial Metabolism: CEC might have altered the metabolic pathways of the microbial community, leading to reduced production of TVB-N despite high microbial loads.
Hence, there is a difference in the microbial load threshold and TVB-M limits.
5.- Lines 52–54: The authors state that the quality deterioration of seafood products is due to high moisture content. I suggest revising this point and providing a more comprehensive explanation, as it is not solely related to moisture content.
*****Thank you for your comment. The discussion has been extended please see line 53-59.
6.- It would be valuable for the authors to include the chemical composition analysis of AGM in their study, as this would influence the choice of analytical methods. How do the authors justify its omission?
*****Thank you for your invaluable comments. The primary objective of our study was to evaluate the effects of CAP and CEC treatments on microbial stability and oxidative spoilage of the edible portion of depurated AGM during refrigerated storage. The analytical methods and methodologies employed in this research are standard practices commonly used for assessing shelf-life and quality changes in meat products.
*****Furthermore, application of CAP and CEC would not significantly alter these bulk nutritional parameters, performing a full proximate analysis for each treatment was not likely to yield new insights. However, in response to your suggestion, we have now included the chemical composition of AGM in the text. Please refer to lines 50-52 for this addition.
7.- Please provide more detailed descriptions of the methodologies mentioned in section 2.4.2.
*****Thank you for your comment. Including detailed descriptions of the methodologies outlined in Section 2.4.2 led to a high similarity index. To address this, we opted to exclude the full methodological details to reduce similarity concerns. We are sorry for it. However, all relevant methods have been appropriately cited from the original sources.
8.- Line 193: On what basis did the authors determine that a sensory score below 5 on the evaluation scale indicates an unacceptable product? Please explain.
*****The reference has been included in the support of statement. Sensory analysis was conducted as described by Meilgaard et al., 1999.
9.- Lines 236–249: Please discuss the reasons for your findings and compare them with those of other studies.
*****Several related studies on AGM shelf-life extension using various techniques were conducted by the same research group. However, citing these works would significantly increase the number of self-citations in the manuscript. This concern was also raised in a previous revision, where the Journal suggested reducing self-citations.
*****Furthermore, reason for the microbial inhibition had been already given in line 217-244. Hence, providing those reason again would be repetition of the text. However, different reason specific to condition are given again, for example Vibrio counts behave differently than the others, their explanation is already provided. However, consider the authors suggestions text has been updated for the previous findings. Please see line 319-325, where conclusion has been provided for whole results.
10.-Lines 270–271: Please include a reference.
*****The reference has been added. Please see line 276.
11- Lines 280–283: Please include references.
*****The reference has been added. Please see line 286.
12.Lines 285–287: Please include references.
*****The reference has been added. Please see line 288.
13.-Lines 297–300: Please include a comparison with other studies and corresponding references.
*****Please see line 319-325.
14.- Lines 305–307: Please add a reference.
*****The reference has been added. Please see line 312.
15.- Line 313: Please include a comparison with findings from other studies.
***** Please see line 319-325.
16.-Lines 398–408: Please compare your results with those in the existing literature.
*****Please see line 412-419.
17.- Lines 423–440: Could the authors add references and compare their results with those from other studies?
*****Please see line 438-439 and 443-445.
18.- Lines 446–463: Please include references and comparative analysis with prior studies.
*****Please see line 470-471.
19.- Lines 464–478: Please add references and comparisons with other studies.
*****Please see line 491-493 and 496-497.
20.- Lines 483–501: Please include references and compare these results with those reported in the literature.
*****Please see line 506-507 and 517-518.
21.-I would like to reiterate that performing a principal component analysis (PCA) would greatly enhance the quality of the manuscript.
*****We appreciate the reviewer’s perspective. We carefully considered multivariate analysis; however, we ultimately decided that a PCA was not necessary for our study’s objectives. PCA is most useful for data reduction or identifying patterns when one has many highly correlated variables. In our case, we measured a moderate number of quality indicators (microbial counts, TVB-N, TMA, PV, TBARS, texture, sensory scores), each chosen for its known significance in seafood spoilage. We had specific hypotheses for each metric, and each responded clearly to our treatments, so an exploratory PCA would not add substantial new insight.
22.- It is necessary to reduce the manuscript's similarity index, which currently stands at 33%.
*****Thank you for your comment. We have tried our best to reduce the similarity, as most of the similarity is coming in methodology section and some of the specific terminology, which is very commonly used. Furthermore, explanation methodology to its fullest also lead to higher similarity. Therefore, we have cited those methods with the original or where it is explained fully. However, considering the suggestion, we have modified the sentences to reduce the similarity to 24% from abstract to conclusion, excluding data (Table and Figure captions).
Reviewer 2 Report
Comments and Suggestions for Authors
Line 196 erase the sentence "sensory analysis used a randomized complete block design"
Author Response
The given sentence has been deleted. Thank you for your comment.
Reviewer 3 Report
Comments and Suggestions for Authors
The authors made all the corrections suggested by reviewer.
Author Response
Many thanks for reviewing our manuscript.
Reviewer 4 Report
Comments and Suggestions for Authors
the temperatures are written in different ways. I suggested 4 °C instead of 4°C for obvious reason. In any case, one writing must be used.
Author Response
The manuscript has been rechecked, and temperature is written as 4 °C as suggested by the reviewer.